# Infrared Thermography in Assessment of Facial Temperature of Racing Sighthound-Type Dogs in Different Environmental Conditions

**DOI:** 10.3390/ani14081180

**Published:** 2024-04-14

**Authors:** Anna Budny-Walczak, Martyna Wilk, Robert Kupczyński

**Affiliations:** 1Department of Environment Hygiene and Animal Welfare, Faculty of Biology and Animal Science, Wroclaw University of Environmental and Life Sciences, 38c Chelmonskiego St., 50-375 Wroclaw, Poland; 2Independent Researcher, 64-320 Niepruszewo, Poland; martyna.wilk@biodose.net

**Keywords:** coursing, heat stress, IRT, sighthound dog, thermoregulation, whippet

## Abstract

**Simple Summary:**

Greyhound welfare science, with a particular emphasis on the heat stress associated with greyhound racing, which often takes place during the summer, is an area of growing interest. Hyperthermia in sporting and working dogs is one of the greatest threats to their health due to physical exercise, often in suboptimal environmental conditions. Monitoring the health of these dogs must be a priority, and for this purpose, it is worth using non-invasive methods that reduce additional stress. One such tool is infrared thermography (IRT).

**Abstract:**

The aim of the study was to assess the usefulness of IRT measurements of selected regions of interest (ROI), i.e., the eyeball and the nose of whippet dogs, before and after coursing competitions taking place in various environmental conditions, thereby enabling the assessment of well-being and the level of heat stress. The research was carried out over two different periods with different thermal humidity indexes (THIs). In the first period, the THI was 59.27 (Run 1), while in the second period, the THI was 63.77 (Run 2). The experimental subjects comprised 111 sighthound-type dogs—whippets—that were photographed with a thermal imaging camera to determine their eye temperature (ET) and nose temperature (NT). The average minimum and maximum eye temperatures were statistically lower after running in both measurements. Increased minimum and maximum nose temperatures were also demonstrated after both runs. The nasal temperature values were statistically higher for Run 2, for which the THI was higher, compared to Run 1. Eyeball temperature may be a marker of thermoregulation ability, regardless of the ambient temperature. The value of ETmax decreased on average by 2.23 °C and 0.4 °C, while NTmax increased uniformly by 2 °C after both runs. A correlation was found between the IRT measurements and physiological indicators.

## 1. Introduction

Greyhound welfare science, with a particular emphasis on the heat stress associated with greyhound racing, which often takes place during the summer, is an area of growing interest. Non-pyrogenic hyperthermia is one of the greatest threats to sporting and working dogs [1]. This condition occurs when a dog’s body temperature rises above 39.4–41 °C due to exercise and/or environmental conditions that impede thermoregulation [1,2]. At environmental temperatures higher than body temperature, evaporation in dogs occurs mainly through panting. This enables effective heat loss due to the large evaporation surface of moist mucous membranes, the mucous membranes of the nasal turbinates, and the oral cavity, also favored by increased salivation [2]. On the other hand, an increase in muscle activity generates a certain amount of heat. Working and sporting dogs may be more susceptible to exertional non-pyrogenic hyperthermia due to increased activity [3]. In the case of sporting dogs, the racing season begins mainly in summer, which is associated with the dogs’ activity in conditions that are unfavorable for maintaining thermal comfort. Hyperthermia in dogs can be divided into three stages: heat stress, which includes initial physiological reactions to increased body temperature; then exhaustion, where mild-to-moderate organ damage occurs; and stroke, which causes neurological symptoms and serious organ damage and may result in death [1]. McNicholl et al. [4] indicate a moderate impact of ambient temperature on the body temperature of greyhounds, whose body temperature can be >41.5 °C. Parnes et al. [3] indicate the multifactorial nature of the possibility of heat stress occurrence and mention, among other things, the impact of workload, environmental conditions, acclimatization, body hydration, and the caregiver’s guidance/reaction, i.e., appropriate cooling methods.

To determine the environmental conditions conducive to the occurrence of heat stress, the thermal humidity index—THI—is used in zootechnical science. The index takes into account the combined effects of relative humidity and ambient temperature. For animals, THI values below <72 are assumed to be thermally comfortable conditions [5,6]. Heat stress also occurs when the heat load is greater than heat loss, so in post-training conditions, an owner should facilitate thermoregulation for their dog.

Infrared thermography (IRT) involves detecting the thermal energy emitted by so-called thermal windows. The level of energy emitted is directly related to blood circulation because the heat generated by the muscles is transferred this way [7]. This technology is non-invasive, so its application has been expanding to other fields of zootechnical and veterinary sciences. In animal studies, research areas were identified and called “thermal windows” due to the presence of a dense network of blood vessels, plexuses, and arteriovenous anastomoses and the lack of hair cover in a given place, which allows for a better thermographic image [8,9]. Changes in the temperature of these areas may indicate various ailments, e.g., stress, injuries, arthritis, fever, and viral infections [8,10]. In a study of greyhounds, scientific reports using thermographic imaging describe changes in the temperature of body areas and limbs during races, which could predict injuries [11]. Other applications include assessments of dog biomechanics, gait analysis [7,12], the identification of musculoskeletal disorders [13,14,15]. Thermography has also been used to examine changes in skin temperature after exercise on a treadmill, indicating a correlation between exercise and skin surface temperature [16,17,18].

A very valuable “measuring window” is the eyeball—the ocular window. Many IRT studies have shown that eyeball temperature has a significant correlation with body temperature and rectal temperature in dogs [19,20], as well as in cattle and sheep [21]. This correlation is attributed to the proximity of the eye socket to the brain as well as its rich blood supply [10]. Eye temperature (ET) is also important in studies of canine stress responses. A decrease in ET may reflect a response to stress dominated by the sympathetic nervous system, while an increase may reflect a response to stress dominated by the parasympathetic nervous system [10].

Another described window is the nasal thermal window. It is an area of particular interest in research on exotic animals due to the ability of the sympathetic branch to reflect the activity of the autonomic nervous system during stressful events that cause emotional arousal, physical activity, and disease processes, as they cause peripheral vasoconstriction [9]. Research on primates has shown that this region is valuable for determining signs of stress [9]. The nasal surface temperature was higher in individuals showing excitement, i.e., in the presence of food, while it decreased in negative interactions [9].

The aim of the study was to assess the usefulness of IRT measurements of selected regions of interest (ROI), i.e., the ET and the NT, of whippet dogs before and after coursing competitions taking place in various environmental conditions determined by the THI. The research assessed the usefulness of using IRT compared to standard physiological measurements when assessing the possibility of thermal stress in dogs and the effectiveness of thermoregulation. We hypothesized that the measurement of the temperature of the eyeball and nose using thermographic imaging would change significantly after exercise, enabling the assessment of well-being and the level of heat stress.

## 2. Materials and Methods

### 2.1. Animals

The research was conducted on 111 whippet dogs during coursing competitions on two dates: Run 1, April 2023, Konstantynów Łódzki, Poland (51 dogs); Run 2, July 2022, Łowicz, Poland (60 dogs). The dogs were 2–5 years old, their weight was 11–16 kg, and the gender distribution was even in each race. The tests were conducted with the consent of the dog’s owner and were completely non-invasive.

The physiological parameters analyzed were body weight (BW), rectal temperature (RT), oxygen saturation (SpO_2_), pulse rate (PR), and respiratory rate per minute (RR). Body weight was provided by the caregiver after examination in a veterinary clinic. RT, SpO_2_, and PR parameters were measured using the T4NT Vital Signs Monitor (TOOTOO Meditech Co., Ltd., Shenzhen, China). Canine pulse oximetry probe placement was on a nonpigmented area of the lip. The respiratory rate was calculated as the number of times the dog’s chest rose per minute. The physiological parameters were resting parameters determined for the same dogs one hour before the race. An assessment of these parameters after the race was not possible due to organizational reasons.

Before the competition, the dogs underwent veterinary examinations. A veterinarian was also constantly present during the competition. Coursing races in Poland take place in pairs and follow a 550 m route with turns which the dogs must overcome behind a lure led at a speed adapted to the given trampling pair.

The study was carried out in accordance with current European legislation (directive 2010/63/UE), and all experimental procedures associated with animal use in the study were approved by the Animal Care Committee of the Wrocław University of Environmental and Life Sciences. The distance from the dog when taking photos was not a stressful element, so this was not factored into the analysis. The studies measured environmental indicators.

### 2.2. Environmental Conditions and IRT

Ambient temperature (AT) and relative humidity (RH) were measured every 2 h using a thermohygrometer Multifunctional Environmental Measurement Device 4 in 1 Voltcraft^®^ (Conrad Eletronic SE, Hirschau, Germany). The thermal humidity index (THI) was calculated by using the following formula [22]:THI = (1.8AT + 32) − (0.55 − 0.0055RH/100) × (1.8AT − 26)(1)
where

THI—temperature humidity index;RH—relative humidity;AT—ambient temperature.

Ground temperature (GT) was also measured using a thermal imaging camera, the Flir^®^ T540 (Teledyne FLIR, Wilsonville, OR, USA).

All thermal images were taken with a thermal imaging camera, the Flir^®^ T540, with a resolution of 464 × 348 (Teledyne FLIR, Wilsonville, OR, USA). The thermal images included the eyeballs (ET) and nose (NT). The photos were taken just before and after the start at a distance of approx. 60–100 cm (a 40 cm difference will not affect the measurement due to the high resolution of the equipment). The emissivity was set at 0.98 based on the literature [20].

The FLIR Thermal Studio program was used to read the average ROI temperature, i.e., the ET of the left and right eyes and the NT. Measurements were made for the medial angle of the eye, the lacrimal caruncle (LC), which was also the value of the maximum temperature, ETmax, and the minimum temperature, ETmin, of the eyeball, which was the lacrimal gland (LG) region. For the temperature of the nasal surface, the maximum temperature, NTmax, and the minimum temperature, NTmin, were read. An example IRT image is shown in Figure 1 with the research regions marked.

### 2.3. Statistical Analyses

Descriptive statistics were analyzed as the mean value and standard deviation (SD), as well as the standard error of the mean (SEM), for normally distributed variables. Analyses of baseline variables were performed between groups using commercially available statistical software, Statitica 13^®^ (StatSoft Inc., Tulsa, OK, USA). To evaluate the differences between the IRT mean values of independent samples for Run 1 and Run 2 (separately for mean values before and after a run) and the values of physiological parameters (RT, SpO_2_, PR, and RR) before a run, a Student’s *t*-test was used. For the mean values of the IRT parameters before and after a run, a paired sample *t*-test was used. All analyses were considered significant if *p* < 0.05. In the present study, a Spearman correlation heatmap for the partial significant indicators detected was generated to examine the correlations between the ET (min and max left; min and max right) and the NT (min and max) before a run and the physiological parameters measured before a run: BW, RT, SpO_2_, and PR, RR.

## 3. Results

The ground temperature (GT), ambient temperature (AT), relative humidity (RH), and thermal humidity index (THI) between Run 1 and Run 2 are presented in Table 1.

The average values of the RT and physiological parameters measured before the run as resting parameters are presented in Table 2. No statistically significant differences were noted.

The average minimum and maximum temperatures obtained from the IRT images of the left eye, right eye, and nose before and after Run 1 and Run 2 are presented in Table 3, along with the results of the statistical analysis. During Run 1, there were statistical differences at the *p* < 0.01 level between all the parameters before and after the run. During Run 2, differences at the *p* < 0.05 level were obtained for the left eye (ETmin and ETmax) before and after the run. The right eye minimum temperature (ETmin right) and nose temperature, both the min and max, showed differences at the *p* < 0.01 level before and after the run. No difference between before and after the run was obtained for the right eye maximal temperature (ETmax right). The differences between the runs (Run 1 and Run 2) were statistically significant at the *p* < 0.05 level for the minimum values of ET right, ET left, and the NT and at the *p* < 0.01 level for the maximum values of ET right, ET left, and the NT. There was no difference found between ETmin and max for the left as well as the right eyeball during the same run (Runs 1 and 2) or between ETmin and max for the left and right eye after Run 1 and NTmin and max after Run 2.

During the observations, no case of heat stroke was recorded, but during Run 2, after completing the run, the dogs showed normal symptoms of thermal stress, i.e., increased BP and salivation, which gradually calmed down after appropriate care by the guardian. The post-race care provided by the dog’s guardian consisted of taking off the racing clothes, providing drinking water, a calm trot in a shaded place, and then resting in an air-conditioned car. Some caregivers also had ice gel packs for cooling.

The correlation between the IRT results and physiological parameters is presented in Figure 2. A positive, moderate correlation (0.20 < r < 0.80) was found between ETmin and ETmax, as well as between the left and right eyes. A similar relationship was demonstrated for NTmin and NTmax and ETmin and ETmax for the left eye, but a negative correlation, r = −0.40, for NTmin, NTmax, and ETmin was obtained for the right eye. There was a strong negative correlation, r = −0.80, between the RT and ET for the left eye. There was also a strong negative correlation, r = −0.80, between the RR and the ETmax of the left eye and the ETmin of the right eye, but no correlation between the RR and the ETmax of the right eye. No correlation was found between ETmax for the left and right eyes and the PR. The value of NT was positively correlated, r = 0.20, with SpO_2_ and PR, RT, and RR.

## 4. Discussion

Infrared thermography can be a useful method for inferring heat stress conditions in sporting dogs. Resting physiological parameters indicate the normal condition of dogs and reflect the correctness of the training. In our own research, we found a greater importance to using IRT data such as the ETmax of the eyeball after running than nasal temperature. The increase in ambient temperature after an activity like running may disturb the thermal balance of the body, which results in the eyeball being maintained at a constant high temperature.

The thermal humidity index is an index indicating the possibility of heat stress. The threshold for animals was set at THI = 72. In the present study, during both runs, the THI did not exceed the mentioned threshold, indicating thermal comfort conditions. However, during Run 2, a high RH was noted, which may have hindered thermoregulation through physiological evaporation. The ground temperature during Run 2 was also significantly higher than during Run 1. Azeez et al. [5] describe the reaction of dogs to conditions where the THI = 80.5–85.5, which showed symptoms of thermal stress by increasing the RR and heart rate. In conditions with a higher RH, the authors noted statistically higher concentrations of cortisol in the blood [5]. The resting physiological parameters in our own study indicate the dogs’ normal conditions and reflect the correctness of the training before the runs. Zanusso et al. [23] showed similar results of SpO2 ranges of 93–100, as well as a similar value of PR, 95 ± 29 bpm. Lopedote et al. [24] obtained results higher than those shown in the manuscript. They were, respectively, a PR of 106 ± 16 bpm and an RR of 54.12 ± 19.78 breaths/min. The difference is insignificant for the PR, while the higher RR value found by Lopedote et al. [24] may be due to the dogs’ level of excitement before the field trial. This theory can be confirmed by the lower value measured at home, which was 26 ± 7.21 breaths/min. The RT found by Zanusso et al. [23] was 36.4 ± 1.2 °C and concerned dogs of different breeds, whereas Lopedote et al. [24] examined the RT in working dogs at rest right before a run, and the mean value was similar to those obtained in our own study (RT 38.3 °C/37.84 and 37.93 °C). 

In the current study, no significant differences were found between the temperature of the right and left eyes before and after the race, which is confirmed by other studies in dogs and dairy cows [16,25]. The average minimum temperature for the eye region was statistically lower after running for both runs. The maximum temperature of the right eye did not show statistically significant differences before and after the run in the environmental conditions prevailing during Run 2. However, during Run 1, the maximum temperature of the right eye showed statistically significant differences before and after the run—34.83 °C and 32.69 °C (*p*-value = 0.001). These results differ from those obtained by Elias et al. [10], who showed a higher ET after the race than before. A higher temperature after running in conditions of a THI > 80 may be related to physical exercise and exercise hyperthermia, which was not observed in the dogs in our research at THIs 59.27 and 63.77. The aspect of higher THI values needs to be assessed. Authors conducting research on treadmills have shown that the ET will increase as dogs exercise and consume energy [16]. A decreased ET may reflect a response to stress dominated by the sympathetic nervous system and vasoconstriction [10]. Whereas studies in cattle suggest that the ET is reduced when animals are appropriately startled or in pain [26]. A study in dogs showed that ear temperature decreased in dogs with separation anxiety [27]. Increased eye temperature may also reflect an emotional reaction to dog racing, while in Poland, coursing competitions are treated more as a hobby, and the dogs did not show extreme emotions related to the start. Dogs competing in coursing competitions in Poland are kept in the shade or in air-conditioned cars until the start, which may also account for differences in the compared reports, in which we do not learn about the dogs’ conditions before the start. The thermal conditions shortly before a run could also have influenced the results.

The highest ETmax in each case was in the lacrimal caruncle, while the minimum ETmin was in the lacrimal gland region. The LC temperature is confirmed by the current literature [10,28]. The phenomenon is explained by the deposit of capillaries surrounding the posterior edge of the eyelid and the lacrimal caruncle [10]. The LG temperature has not been precisely described in the literature. The canine lacrimal gland (LG) is responsible for producing the aqueous part of the precorneal tear film, which may influence the temperature of a given ROI [29]. Eilas et al. [10] emphasize the importance of the LC due to its affinity for reflecting stress reactions and pain. Nevertheless, the authors believe that the LG may also provide important information about a dog’s well-being after a race because, in high RH conditions (Run 2), the ETmin measurement showed statistically significant differences between the ‘before run’ and ‘after run’ results.

Relative humidity had a significant effect on ETmin and ETmax, which were lower after the races than before, while during Run 2 (RH = 72%), the differences were lower than during Run 1 (RH = 42%). It is possible that this was due to the fact that humidity may disturb thermoregulation in dogs, especially those with greater activity [4,10,30], who, relying on latent evaporation, are unable to release excess heat to an environment with a low humidity deficit.

The minimum and maximum temperatures of the external nasal mucosa were statistically different before and after the run for both runs, and they reached a higher value after the runs. The nose temperature values were statistically higher for Run 2, for which the THI was 63.77, compared to Run 1, for which the THI was 59.27. However, the differences in NTmax before and after Run 1 and Run 2 were not significantly differentand amounted to 4.99 and 5.08 °C, which would indicate the lack of response of this region to a higher RH. The nasal window is an area that has the ability to reflect the activity of the autonomic nervous system during stressful events that cause emotional arousal, physical activity, and disease processes because these processes, through secreted hormones, cause peripheral vasoconstriction [9]. Kennedy et al. [31] did not demonstrate a relationship between nasal temperature and rectal temperature in dogs. The values of the NTs of the tested dogs obtained by Kennedy et al. [31] were 17.8–38.3 °C, which can be compared to the results obtained in the presented study for both runs. Studies on primates have shown that a high NT reflects signs of excitement [9]. After finishing the runs, the dogs showed higher NTs, which can be explained by the effort and emotions associated with the runs, while NTmin and max were statistically higher in conditions of an increased THI, i.e., Run 2. This fact should be noted as an influence of the ambient temperature on the temperature of the nasal mucosa.

The correlation between the IRT results and physiological parameters was moderate to strong but confirms the validity of using the thermographic imaging method in monitoring the well-being of sporting dogs. Strong negative correlations between the RT and ET for the left eye and between the RR and ETmax for the left eye and ETmin for the right eye support our statement.

During both runs, none of the dogs showed symptoms of stroke, only mild heat stress, i.e., panting and salivation after the run [2,32]. A few dogs during Run 2 showed slight signs of heat stress before the start, but their condition did not deteriorate after the start. Greyhound dogs are bred mainly for racing, and selective breeding influences the properties of these dogs and their anatomy, which favors thermoregulation mechanisms. The thermoregulation mechanisms of dogs during a race involve convection, radiation, and physiological evaporation from the skin surface and the upper respiratory tract. Running speed causes air movement and forces convection. The dilation of peripheral vessels caused by activity also facilitates the release of heat through radiation. The anatomical structure of greyhounds, which is characterized by high musculature and a small amount of fat tissue, as well as a thin layer of coat, also contributes to heat loss through radiation and convection due to poor heat insulation. The methods of thermoregulation of whippet dogs while running seem to be sufficient for the described environmental conditions.

The results of the present study suggest that the ambient temperature has an impact on the temperature of the eyes and nose, which indicates the validity of using the thermographic imaging method in the future to monitor the well-being of racing greyhounds related to heat balance and predict the occurrence of heat stress.

## 5. Conclusions

Infrared thermography is a useful method for inferring states of heat stress and emotional arousal for the welfare of sporting dogs. This method can be used to measure the temperature of the eyes and nose to determine their response to environmental and stress factors. An additional problem is the fact that dog racing often takes place in the summer season, which makes the animals’ thermoregulation difficult. Eyeball temperature can be a marker of thermoregulation ability, regardless of the ambient temperature. Nasal temperature after running reflects environmental conditions to a lesser extent. Significant correlations were noted between the temperature of the eyes (max and min) and the temperature of the nose (max and min), which proves their interdependence and the importance of examining these parameters together. The positive correlation between the IRT parameters and saturation, as well as the negative correlation between these parameters and the RT, PR, and RR, provide a basis for assessing clinical symptoms of heat stress. The measurement of RT, SpO2, PR, and RR and thermal imaging measurements can be helpful tools in heat stress diagnosis in performance dogs and also in evaluating the effectiveness of cooling methods. Future research should be devoted to these issues. 

To sum up, the thermographic imaging of whippet sports dogs is a good tool for assessing the dogs’ well-being related to physical exercise and temperature changes at points known as thermal windows.

## Figures and Tables

**Figure 1 animals-14-01180-f001:**
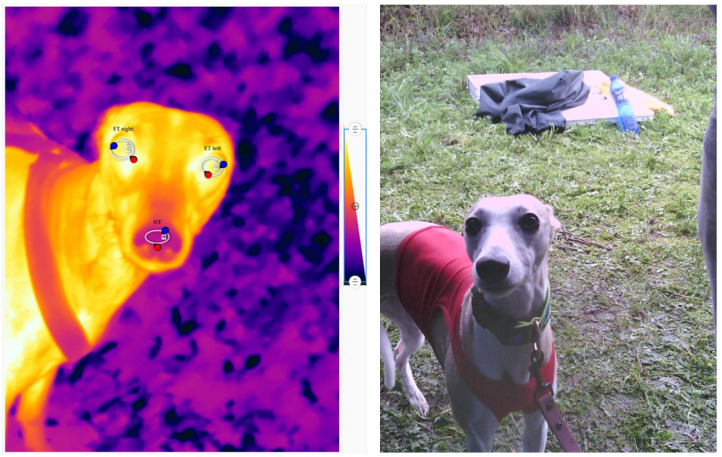
IRT image with marked ROI of observation of left eye (ET left), right eye (ET right), and nose (NT). Red dots are showing the maximal temperature, and blue dots are showing the minimal temperature of the marked regions.

**Figure 2 animals-14-01180-f002:**
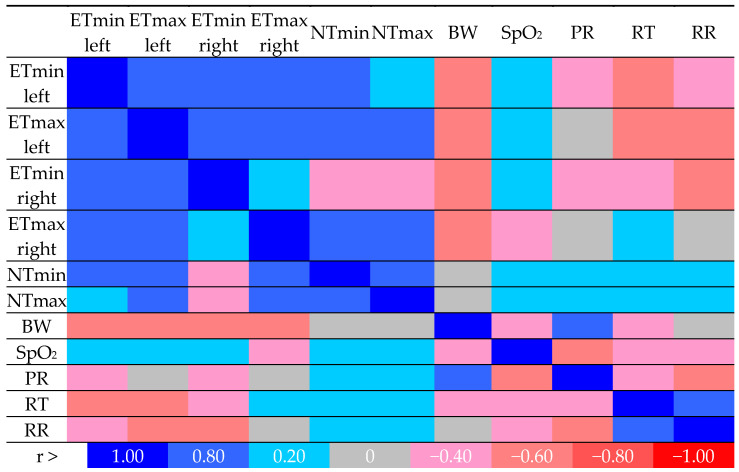
Spearman correlation matrix with r values from log2 fold change values of IRT parameters (ETmin and max, left and right—eye temperature; NTmin and max—nose temperature) and physiological parameters (BW—body weight, RT—rectal temperature, SpO_2_—oxygen saturation, PR—pulse rate, RR—respiratory rate). Legend is presented in the last line. The red color indicates a strong negative correlation, the blue color indicates a strong positive correlation, and the gray color indicates no correlation.

**Table 1 animals-14-01180-t001:** Ground temperature (GT), ambient temperature (AT), relative humidity (RH), and thermal-humidity index (THI) during Run 1 and Run 2.

	GT [°C]	AT [°C]	RH [%]	THI
Run 1	11.00	16.00	42	59.27
Run 2	23.70	21.50	72	63.77

**Table 2 animals-14-01180-t002:** Average values of rectal temperature (RT), oxygen saturation (SpO_2_), pulse rate (PR), and respiratory rate (RR) before the race.

	RT [°C]	SpO_2_ [%]	PR [bpm]	RR [Breaths/min]
Run 1 *	37.84 ± 0.54	94.38 ± 4.38	94.46 ± 16.15	38.39 ± 13.80
Run 2 *	37.93 ± 0.41	95.35 ± 3.53	99.56 ± 17.58	42.63 ± 12.88
*p*-Value	0.19	0.29	0.68	0.77

* Rest parameters.

**Table 3 animals-14-01180-t003:** Mean minimum (min) and maximum (max) temperatures of examined regions obtained from IRT images.

ROI		Before Run		After Run		*p*-Value
		Mean ± SD	SEM	Mean ± SD	SEM	
Run 1						
left eye	ETmin	30.87 ± 1.86 ^a^	0.27	28.56 ± 2.24 ^a^	0.45	<0.001
	ETmax	34.87 ± 1.11 ^A^	0.16	32.54 ± 1.83 ^A^	0.37	<0.001
right eye	ETmin	30.24 ± 1.83 ^a^	0.29	27.81 ± 3.24 ^a^	0.76	0.001
	ETmax	34.83 ± 0.83 ^A^	0.13	32.69 ± 1.88 ^A^	0.44	0.001
nose	NTmin	17.81 ± 3.03 ^b^	0.44	20.60 ± 4.35 ^b^	0.85	<0.001
	NTmax	21.23 ± 3.57 ^B^	0.52	26.22 ± 3.99 ^B^	0.78	<0.001
Run 2						
left eye	ETmin	32.50 ± 1.15 ^c^	0.18	31.54 ± 1.72 ^c^	0.28	0.015
	ETmax	35.50 ± 0.78 ^C^	0.12	35.10 ± 1.03 ^C^	0.17	0.014
right eye	ETmin	32.45 ± 1.16 ^c^	0.18	31.31 ± 1.58 ^c^	0.26	0.004
	ETmax	35.49 ± 0.70 ^C^	0.11	35.18 ± 1.02 ^C^	0.17	0.194
nose	NTmin	24.06 ± 2.73 ^d^	0.39	28.48 ± 2.36 ^a^	0.36	<0.001
	NTmax	26.02 ± 3.09 ^D^	0.45	31.10 ± 2.25 ^A^	0.34	<0.001

^a–d^ Statistical differences denoted by different lowercase letters are at the *p* < 0.05 level for the minimum (min) temperature between regions (column). ^A–D^ Statistical differences by different uppercase letters are at *p* < 0.01 level for the maximum (max) temperature between regions (column). ROI—region of interest; ETmin—minimal eye temperature; ETmax—maximal eye temperature; NTmin—minimal nose temperature; NTmax—maximal nose temperature; SD—standard deviation; SEM—standard error of the mean.

## Data Availability

The data presented in this study are available on request from the corresponding author.

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
