# Peer review of "Infrared Thermography in Assessment of Facial Temperature of Racing Sighthound-Type Dogs in Different Environmental Conditions"

_animals, 2024, doi:10.3390/ani14081180_

Round 1

Reviewer 1 Report

Comments and Suggestions for Authors

The authors described the feasibility of IRT as dog health and stress monitoring. This manuscript focused on the whippet that participated in the competition race. I was impressed by the results of the IRT differences between LC and ET. Further survey is needed to generalize these methods, but this is the point at which to start revealing the mechanism of IRT on the dog’s face.

Minor revision

I found a mistake on Line 174 that should be translated!

The other sentences are fine.

Author Response

Dear Reviewer,

Thank you for your valuable comments regarding manuscript ID: animals-2913579, entitled IRT in assessment of thermoregulation of whippets during coursing competitions in different environmental conditions. We appreciate your detailed review and hope that our statements will find your acceptance.

Reviewer 1

Line 174: I found a mistake on Line 174 that should be translated!

AU: Translation done: „No correlation was found between ETmax left and right eye and PR”. (added in line 192)

Reviewer 2 Report

Comments and Suggestions for Authors

Overall this is an interesting study of an important topic and it is moderately well-written.

  One notable flaw is that after providing results which often differed from previous studies, (i.e. correlation between NT and RT where previous study showed no correlation, and lowered ET whereas previous study showed increased ET after running.) that the authors conclude that "IRT is a good tool for assessing the dogs' well-being..."  (Lines 262-264; 275-277).   If the results are not repeatable between studies, then the accuracy and usefulness of that tool can't be yet confirmed.   I feel the authors need to discuss in more detail the differences between theirs and previous studies, and try to give a more in-depth idea of why these might have differed than in previous studies other than the vague concept that because these dogs are competing on a "hobby level" they might not have been as stressed. (Lines 204-207).  If that is the case, how could you test this idea?  What other factors could have affected your results that made them different to previous studies?  Environment?  Breed?  Technical methods of use of the IRT camera?

Also, this study did not actually test the ability of IRT to assess well-being, as the state of "well-being" was not assessed nor compared amongst these dogs.  Rather, it studied the ability of IRT to correlate to other parameters such as rectal temperature, heart rate, and respiratory rate is dogs that, as a population, showed no signs of clinical heat injury following exercise.  Multiple studies in dogs over several decades now have shown that body temperature and other vital signs are not accurate in indicating or predicting heat injury in dogs as none of the dogs in these studies on exercise hyperthermia resulted in dogs becoming heat casualties even at temperatures as high as 41 C.  The severity of neurologic and other systemic factors are used to categorize the severity of heat injury rather than the body temperature.  So, now that you have the ET and NT, what does that actually mean to the dog's immediate clinical state or well-being?

I believe the conclusions would be more accurate to simply state what parameters showed positive or negative correlations and conclude whether IRT measurements of the selected anatomical windows of eyeball and nose are repeatable and consistent amongst their population and with previous studies, and based on those findings, comment on their potential to be useful in a practical way to assess clinical signs of heat stress. I don't think at this time a conclusion that this is a useful tool to assess state of health is accurate.

Lines 103-104: Other comments/Questions:  Body weight was provided by the caregivers.  How did you ensure that these BW values were accurate enough to be included in your analysis?  Where and how did the caregivers measure the body weights?

130-131: What effect does distance between the measured surface and the camera have on reading?  Your distances varied within a range of 40cm.  I am aware that higher resolution cameras can be more accurate over varying distances, but please comment on this in your materials and methods and/or discussion as previous studies have typically standardized the distance within a few centimeters between readings. 

Line 156: Regarding resolution of post-run signs of mild thermal stress, what was "appropriate care by the guardian?"  Were specific methods observed that authors could comment on?  (Likely a variety of methods between caregivers, but would like to see this described a bit more thoroughly.)

Line 173: Brief sentence not in English like the rest of the manuscript.

Lines 195-197: "...showed statistically significant differences..."  please state what those differences were so the reader does not have to search back through the results to find them.

Lines 196-197: "These results differed from those obtained by Azeez et al (4) who showed higher ET after the race than before."   Reading the Azeez article as cited, it does not look like Azeez et al studied dog tempratures before and after runs, and did not utilize IRT in that study.  Perhaps this is just inadvertently citing the wrong study?

Whichever study that was intended to cite, why do you think your study results differed from previous studies?  For example, the increased vs decreased ET in previous study compared to your study, and no correlation in of NT and RT in the Kennedy study but positive correlation in yours?  I think you need to give more in-depth consideration to why your results may have differed, and how this effects the assessment of accuracy of IRT in measurement or assessment of canine thermoregulation.

Line 198-199: "Higher temperature after running in conditions THI>80 may be related to physical exertion and exercise hypothermia.  Was this meant to say "hyperthermia?"

Line 238-239:  The Kennedy study as well as your study showed such a wide range of nasal temperatures (about 20 C) that it is hard for me to see any relevant or accurate clinical use yet for this measurement in assessing thermoregulatory state or health of dogs.  In fact, the Kennedy study showed no correlation between NT and RT.  I believe that further study may eventually lead to better understanding of thermoregulatory processes and heat compartmentalization in dogs, but is not a practical measurement at this time because we do not have good understanding of what leads to this wide variation in nasal temperature or what it means for health of the dog.

Comments on the Quality of English Language

Use of the English language was very good with some very minor exceptions. 

Author Response

Dear Reviewer,

Thank you for your valuable comments regarding manuscript ID: animals-2913579, entitled IRT in assessment of thermoregulation of whippets during coursing competitions in different environmental conditions. We appreciate your detailed review and hope that our statements will find your acceptance.

Reviewer 2

One notable flaw is that after providing results which often differed from previous studies, (i.e. correlation between NT and RT where previous study showed no correlation, and lowered ET whereas previous study showed increased ET after running.) that the authors conclude that "IRT is a good tool for assessing the dogs' well-being..." (Lines 262-264; 275-277). If the results are not repeatable between studies, then the accuracy and usefulness of that tool can't be yet confirmed. I feel the authors need to discuss in more detail the differences between theirs and previous studies, and try to give a more in-depth idea of why these might have differed than in previous studies other than the vague concept that because these dogs are competing on a "hobby level" they might not have been as stressed. (Lines 204-207). If that is the case, how could you test this idea? What other factors could have affected your results that made them different to previous studies?  Environment?  Breed? Technical methods of use of the IRT camera?

AU: We appreciate your suggestions to improve the quality of the manuscript. Additional aspects of the discussion have been added to the manuscript. Line 230-237, 243-246.

Also, this study did not actually test the ability of IRT to assess well-being, as the state of "well-being" was not assessed nor compared amongst these dogs.  Rather, it studied the ability of IRT to correlate to other parameters such as rectal temperature, heart rate, and respiratory rate is dogs that, as a population, showed no signs of clinical heat injury following exercise.  Multiple studies in dogs over several decades now have shown that body temperature and other vital signs are not accurate in indicating or predicting heat injury in dogs as none of the dogs in these studies on exercise hyperthermia resulted in dogs becoming heat casualties even at temperatures as high as 41 C.  The severity of neurologic and other systemic factors are used to categorize the severity of heat injury rather than the body temperature.  So, now that you have the ET and NT, what does that actually mean to the dog's immediate clinical state or well-being?

AU: We appreciate your suggestions. We believe that by examining the temperature of selected ROIs and simultaneously observing the dog, we can predict the occurrence of heat stress. Other studies with increased THI indicate higher values of eyeball temperature. We may be able to explore the relationship further. Measuring rectal temperature does not provide an effective assessment of thermal stress, but the temperature of the eyeball and nose related to the network of blood vessels and the nervous system may give us a different view of the problem. Unfortunately, I disagree that the work is not about animal welfare. Conditions of thermal comfort and maintaining this state is a welfare issue. Especially when it comes to sports dogs competing in conditions unfavorable for thermoregulation.

I believe the conclusions would be more accurate to simply state what parameters showed positive or negative correlations and conclude whether IRT measurements of the selected anatomical windows of eyeball and nose are repeatable and consistent amongst their population and with previous studies, and based on those findings, comment on their potential to be useful in a practical way to assess clinical signs of heat stress. I don't think at this time a conclusion that this is a useful tool to assess state of health is accurate.

AU: We add sentence according to your suggestion line: 314-319.

Lines 103-104: Other comments/Questions:  Body weight was provided by the caregivers.  How did you ensure that these BW values were accurate enough to be included in your analysis?  Where and how did the caregivers measure the body weights?

AU: We add sentence according to your suggestion line: 110. The body weight was provided by the caregivers who check the given parameter at the veterinarian on a professional platform scale.

130-131: What effect does distance between the measured surface and the camera have on reading?  Your distances varied within a range of 40cm.  I am aware that higher resolution cameras can be more accurate over varying distances, but please comment on this in your materials and methods and/or discussion as previous studies have typically standardized the distance within a few centimeters between readings. 

AU: We add sentence according to your suggestion line: 137-139. Polish research on the influence of the temperature of a given object examined with a thermal imaging camera (336 × 256 resolution) and the distance from it indicates that the standard deviation of the temperature measurement result at a distance from 1 m to 56 m in the case of average temperature is 1.01 (Kowalski and Smyk, 2020 , DOI 10.21008/j.1897-0737.2020.104.0010). This is a small difference at such a large distance, so we can safely conclude that 40 cm will not cause differences in readings.

Line 156: Regarding resolution of post-run signs of mild thermal stress, what was "appropriate care by the guardian?"  Were specific methods observed that authors could comment on?  (Likely a variety of methods between caregivers, but would like to see this described a bit more thoroughly.)

AU: We add sentence according to your suggestion line: 171-174.

Line 173: Brief sentence not in English like the rest of the manuscript.

AU: We translate sentence accordig also to advice of Reviewer 1.

Lines 195-197: "...showed statistically significant differences..."  please state what those differences were so the reader does not have to search back through the results to find them.

AU: We add sentence according to your suggestion line: 230-231.

Lines 196-197: "These results differed from those obtained by Azeez et al (4) who showed higher ET after the race than before."   Reading the Azeez article as cited, it does not look like Azeez et al studied dog tempratures before and after runs, and did not utilize IRT in that study.  Perhaps this is just inadvertently citing the wrong study?

Whichever study that was intended to cite, why do you think your study results differed from previous studies?  For example, the increased vs decreased ET in previous study compared to your study, and no correlation in of NT and RT in the Kennedy study but positive correlation in yours?  I think you need to give more in-depth consideration to why your results may have differed, and how this effects the assessment of accuracy of IRT in measurement or assessment of canine thermoregulation.

AU: That's right, we ment to mentioning Elias, B.; Starling, M.; Wilson, B.; McGreevy, P. Influences on Infrared Thermography of the Canine Eye in Relation to the Stress and Arousal of Racing Greyhounds. Animals 2021, 11, 103. https://doi.org/10.3390/ani11010103. The mistake has been corrected. Discussion about differences in results added.

Line 198-199: "Higher temperature after running in conditions THI>80 may be related to physical exertion and exercise hypothermia.  Was this meant to say "hyperthermia?"

AU: Yes we meant „hyperthermia”. The mistake has been corrected.

Line 238-239:  The Kennedy study as well as your study showed such a wide range of nasal temperatures (about 20 C) that it is hard for me to see any relevant or accurate clinical use yet for this measurement in assessing thermoregulatory state or health of dogs.  In fact, the Kennedy study showed no correlation between NT and RT.  I believe that further study may eventually lead to better understanding of thermoregulatory processes and heat compartmentalization in dogs, but is not a practical measurement at this time because we do not have good understanding of what leads to this wide variation in nasal temperature or what it means for health of the dog.

AU: The temperature range in the article concerned all dogs, including those named as "cold", "intermediate" and "warm". We refer to this article in our coverage for runs with THI 59.27 and 63.77 and suggest that the values are comparable. We agree that further study may eventually lead to better understanding of thermoregulatory processes and heat compartmentalization in dogs. From our experience, we conclude that the ambient temperature has an impact on the temperature of the nasal surface, but exercise at different ambient temperatures has a smaller effect, which we describe in the manuscript. To clarify that the values apply to all dogs, we add sentence in Line 277 and 279.

Reviewer 3 Report

Comments and Suggestions for Authors

Firstly, I would like to congratulate the authors of the manuscript on their very interesting study on

“IRT in assessment of thermoregulation of whippets during coursing competitions in different environmental conditions”. There are few minor observations that the authors should address before final submission.

1.            Rephrase title and avoid abbreviations at the start of a sentence.

2.            Add scientific name of the whippets dogs in title

3.            Line 22 rephrase sentence and avoid figures at the start of a sentence.

4.            Define abbreviations at first citation, throughout the manuscript and avoid abbreviations at the start of a sentence.

5.            L 28 Rephrase sentence “ETmax decreased on average by 2.23ËšC an…….

6.            Arrange keywords alphabetically.

7.            L 91-93 Add objective in abstract as well  

8.            L 105 Vital Signs Monitor make and model?

9.            Add reference for methodology respiration rate etc.

10.        L 126-127 Please add make and model of the camera.

11.        Table 2 Explain statistical alphabets and check if they are correctly labelled? In row 1 similar alphabets depicts non-significant relationship however, p-value depicts significant relationship.

12.        There is a need to improve results and discussion section as well.

13.        Authors should get help from academic writer for academic editing before final submission, please.

Best of Luck. 

Comments on the Quality of English Language

Firstly, I would like to congratulate the authors of the manuscript on their very interesting study on

“IRT in assessment of thermoregulation of whippets during coursing competitions in different environmental conditions”. There are few minor observations that the authors should address before final submission.

1.            Rephrase title and avoid abbreviations at the start of a sentence.

2.            Add scientific name of the whippets dogs in title

3.            Line 22 rephrase sentence and avoid figures at the start of a sentence.

4.            Define abbreviations at first citation, throughout the manuscript and avoid abbreviations at the start of a sentence.

5.            L 28 Rephrase sentence “ETmax decreased on average by 2.23ËšC an…….

6.            Arrange keywords alphabetically.

7.            L 91-93 Add objective in abstract as well  

8.            L 105 Vital Signs Monitor make and model?

9.            Add reference for methodology respiration rate etc.

10.        L 126-127 Please add make and model of the camera.

11.        Table 2 Explain statistical alphabets and check if they are correctly labelled? In row 1 similar alphabets depicts non-significant relationship however, p-value depicts significant relationship.

12.        There is a need to improve results and discussion section as well.

13.        Authors should get help from academic writer for academic editing before final submission, please.

Best of Luck. 

Author Response

Dear Reviewer,

Thank you for your valuable comments regarding manuscript ID: animals-2913579, entitled IRT in assessment of thermoregulation of whippets during coursing competitions in different environmental conditions. We appreciate your detailed review and hope that our statements will find your acceptance.

Reviewer 3

Firstly, we would like to thank you for kind words about our manuscript.

  1. Rephrase title and avoid abbreviations at the start of a sentence.

AU: Thank you for your important comment. The title has been changed.

  1. Add scientific name of the whippets dogs in title

AU: Thank you for your important comment. The title has been changed.

  1. Line 22 rephrase sentence and avoid figures at the start of a sentence.

AU: We changed sentence according to your suggestion.

  1. Define abbreviations at first citation, throughout the manuscript and avoid abbreviations at the start of a sentence.

AU: We changed sentences in tekst according to your suggestion.

  1. L 28 Rephrase sentence “ETmax decreased on average by 2.23ËšC an…….

AU: We changed sentence according to your suggestion.

  1. Arrange keywords alphabetically.

AU: We changed keywords according to your suggestion.

  1. L 91-93 Add objective in abstract as well  

AU: We added objective in abstract according to your suggestion.

  1. L 105 Vital Signs Monitor make and model?

AU: The model was T4NT as it stands in manuscript. Other names of model are not given.

  1. Add reference for methodology respiration rate etc.

AU: The methodology of respiration rate is mentioned in lines 112-113.

  1. L 126-127 Please add make and model of the camera.

AU: The model was T540 as it stands in manuscript. Other names of model are not given.

  1. Table 2 Explain statistical alphabets and check if they are correctly labelled? In row 1 similar alphabets depicts non-significant relationship however, p-value depicts significant relationship.

AU: Statistical alphabets are explained under the tabel. Those are differences between regions (vertical values) and p-value shows significant relationship between stage of run – before/after (horizontal values).

  1. There is a need to improve results and discussion section as well.

AU: Results and discussion is improved according to suggestion of Revier 1 and 2. Will be that enough?

  1. Authors should get help from academic writer for academic editing before final submission, please.

AU: AU: Thank you for your suggestion. After the editor's approval, we will refer for academic editing.

Round 2

Reviewer 3 Report

Comments and Suggestions for Authors

Thanks for sparing time in revising the manuscript. The authors have improved the manuscript, however there are still some minor changes that have not been incorporated. 

1. L 111 Authors were previously requested to add Make and Model of the vital sign monitor. The Authors have mentioned the Model name however, the information regarding country and or the company name of the monitor has not been incorporated.  

2. Similar is the case with the FLIR camera complete information is missing.

An example of reference for ease of authors:

Infrared thermal Camera (FLK-Ti25 9 Hz, Fluke Corporation, Everett, WA, USA)

Author Response

Dear Reviewer,

Thank you for your valuable comments regarding manuscript ID: animals-2913579, entitled IRT in assessment of thermoregulation of whippets during coursing competitions in different environmental conditions. We appreciate your detailed review and hope that our statements will find your acceptance.

Thank you for this suggestion and sorry for the misunderstanding. Additional data has been added to the manuscript according to your suggestion, line: 111-112 and 137.